# Genomic Tracking of SARS-CoV-2 Variants in Myanmar

**DOI:** 10.3390/vaccines11010006

**Published:** 2022-12-20

**Authors:** Khine Zaw Oo, Zaw Win Htun, Nay Myo Aung, Ko Ko Win, Kyaw Zawl Linn, Sett Paing Htoo, Phyo Kyaw Aung, Thet Wai Oo, Myo Thiha Zaw, Linn Yuzana Ko, Kyaw Myo Tun, Kyee Myint, Ko Ko Lwin

**Affiliations:** 1Defence Services Medical Research Centre, Biological Toxin Research Department, Nay Pyi Taw 15021, Myanmar; 2Directorate of Medical Services, Nay Pyi Taw 15013, Myanmar

**Keywords:** COVID-19, SARS-CoV-2, sequencing, variant, Myanmar

## Abstract

In December 2019, the COVID-19 disease started in Wuhan, China. The WHO declared a pandemic on 12 March 2020, and the disease started in Myanmar on 23 March 2020. In December 2020, different variants were brought worldwide, threatening global health. To counter those threats, Myanmar started the COVID-19 variant surveillance program in late 2020. Whole genome sequencing was done six times between January 2021 and March 2022. Among them, 83 samples with a PCR threshold cycle of less than 25 were chosen. Then, we used MiSeq FGx for sequencing and Illumina DRAGEN COVIDSeq pipeline, command line interface, GISAID, and MEGA version 7 for data analysis. In January 2021, no variant was detected. The second run, during the rise of cases in June 2021, showed Alpha, Delta, and Kappa variants. The third and the fourth runs in August and December showed only a Delta variant. Omicron and Delta variants were detected during the fifth run in January 2022. The sixth run in March 2022 showed only Omicron BA.2. Amino acid mutation at the receptor binding domain of Spike glycoprotein started since the second run coupling with high transmission, recurrence, and vaccine escape. We also found the mutation at the primer targets used in current RT-PCR platforms, but there was no mutation at the existing antiviral drug targets. The occurrence of multiple variants and mutations claimed vigilance at ports of entry and preparedness for effective control measures. Genomic surveillance with the observation of evolutionary data is required to predict imminent threats of the current disease and diagnose emerging infectious diseases.

## 1. Introduction

In December 2019, COVID-19 disease was first identified in Wuhan, China [1,2]. The WHO declared the pandemic on 11 March 2020. COVID-19 is caused by a Human Coronavirus, SARS-CoV-2. Human coronavirus (HCoV) is a member of the coronavirinae subfamily of the coronaviridae family [3].

In Myanmar, the first wave started with two cases on 23 March 2020. There were only 374 cases and 6 deaths during the first wave, which ended on 16 July [4]. The second wave started on 16 August in Rakhine State, followed by an outbreak in Yangon occurring among 32,351 cases with 765 fatalities. The first wave was comparatively moderate compared to regional peers, but there was a dramatic rise in the death toll during the second wave. Early June 2021 brought the third wave with a massive number of cases. Cases reached 144,157 with a death toll of 3334, which outnumbered the capacity of quarantine centers and hospitals. At the end of the year, the total confirmed cases reached 530,834, with 19,268 deaths. An Omicron variant was first identified on 7 January 2022, which has caused a raised number of cases reaching 613,577 people with 19,434 deaths as of 29 June 2022 [4].

SARS-CoV-2 is a fast-evolving virus because of rapid and massive genetic variations. The WHO classified viruses with Alpha, Beta, Gamma, Delta, and Omicron as the variants of concern (VOC) and Zeta, Eta, and Kappa as the variants of interest (VOI) [5]. Since December 2020, variants have occurred worldwide, which favors the virus to be fitter for transmission. The high transmission rate caused more morbidity and mortality, endangering existing healthcare systems. All these things affected Myanmar; the country started the COVID-19 variant surveillance program in late 2020, and the Defence Services Medical Research Centre started whole genome sequencing in December 2020.

This study used a WGS technique on the clinical samples collected during the waves above. We described the features of the viral genome sequences from these periods, including VOCs, VOIs, and genetic variation at the spike glycoprotein coding region and primer binding sites, and drew the dynamic of viral spread.

## 2. Materials and Methods

We took 159 SARS-CoV-2 PCR-positive samples from clustered cases in cities and border areas during the second, third, and Omicron waves. We used QIAamp Viral RNA Mini Kit (QIAGEN, Hilden, Germany) for RNA extraction, BioFlux SARS-CoV-2 Nucleic Acid Detection Kit (Bioer, Hangzhou, China), and Applied Biosystems StepOnePlus Real-Time PCR System to check Ct values. We chose samples with a Ct value below 25 for sequencing and finally obtained 101 samples.

We amplified the SARS-CoV-2 whole genome according to the Illumina COVIDSeq RUO Assay (Illumina, San Diego, CA, USA) protocol [6]. Briefly, we synthesized cDNA with the included COVIDseq First Strand cDNA synthesis kit, amplified the whole viral genome selectively with Primer Pool 1 and 2, fragmented and tagged with Bead-linked transposome. Following this, we added Illumina Nextera DNA CD indexes to the samples for identification before we pooled, purified, quantified, and normalized them. Finally, the MiSeq FGx sequencing system amplified them to form clusters and sequenced them using sequencing with synthesis (SBS) chemistry.

We used Illumina DRAGEN COVIDSeq Test Pipeline to trim and assemble FASTQ files to get the FASTA format together with variant and lineage reports. The results were validated again by generating FASTA files with SC-2pipe (BioEasy Sdn Bhd, Selangor, Malaysia) in the local workstation and deposited at GISAID and NCBI GenBank. We collected 189 complete genome sequences from GISAID, which had 99.8% sequence homology with our samples. Subsequently, we removed duplicate and incomplete sequences and finally obtained 78 sequences. We performed sequence alignment using Molecular Evolutionary Genetics Analysis (MEGA) software version 7 with Multiple Sequence Comparisons with Log-Expectation (MUSCLE) algorithm [7], neighbor-joining method, and the Kimura 2 model with 1000 bootstraps for phylogenetic analysis.

In order to assess the effect on viral transmission and immune evasion, we observed the Spike glycoprotein coding region if there were identified nucleotide and amino acid mutations. Mutations at ORF1ab, Nucleocapsid (N), and Envelope (E) encoding regions were also assessed, especially at primer binding sites used in PCR assays. We also collected phenotypic data, including disease severity, vaccination status, and traveling history, to correlate with viral genotypes.

## 3. Results

We chose 83 sequences ranging in length from 28,205 to 29,830 nucleotides, and each contains over 99.75% of the genome (Table 1), and 77 homologous sequences from 15 countries. We drew a phylogenetic tree (Figure 1), and our sequences were distributed independently and were similar to strains from Europe and Asia. Our sequences were similar to those from Malaysia during the first run; India, France, South Korea, China, and Romania during the second run; Thailand, India, South Korea, USA, and Kosovo during the third run; Thailand, Malaysia, India, Austria, Germany, France, Brazil, and England during the fourth run; Malaysia, Philippines, South Korea, India, England, and Ecuador during the fifth run; and Thailand, India, England, China, USA, and Australia during the sixth run.

All our samples showed D614G mutation since the first run with additional Q677H mutation at five out of nine samples. No variant of concern (VOC) or variant of interest (VOI) existed. After six months, we did the second run and found seven VOCs (two Alpha and five Delta) and four VOIs (Kappa). Alpha variants showed a N501Y mutation at RBD, while the Delta showed L452R and T478K mutations at RBD with P681H mutation at the furin binding site (FBS). Similar to Delta variants, Kappa showed L452R and P681H mutations, but there was E484Q instead of T478K. The third and fourth runs had 15 and 16 complete sequences, showing only the Delta variant with the same mutation. Four Delta remained present during the fifth run, together with five Omicron variants. The sixth run in March 2022, with 23 samples, showed only Omicron, all having multiple mutations at RBD and FBS.

Three out of four Delta variants during the fifth run (DSMRC 054, 058, 060) showed Nucleocapsid G28881T, and DSMRC 058, 060 showed ORF1ab G15451A nucleotide substitutions. Three Omicron variants (DSMRC 063, 064, 067) showed trinucleotide mutations (G28881A, G28882A, G28883C) at the Nucleocapsid coding region and Envelope C26270T mutation. All our sequences belonged to non-variant and four variants as well as 15 strains. Generally, they were distributed equally with each other and worldwide sequences. However, the third and fourth runs with the Delta variant became more homologous.

## 4. Discussion

Many variants were found in this study and were similar to those found in India, in England, and worldwide. We found similar sequences from Asia and worldwide without any from Africa and South America. Since late 2021, the Delta variant dominated until the appearance of Omicron in January 2022. The Delta variant took over other global variances after its emergence in India in December 2020 [8].

Phylogenetic analysis showed that our sequences were distributed independently and clustered together with those from 21 countries. These results showed complex geographical and temporal transmission patterns and a high virus mutation rate. Our sequences became more homologous with worldwide sequences during the third and fourth runs. Clustering with those from Europe (France, England) showed rapid disease transmission in today’s world. Clustering with other Asian countries (India, Malaysia, and Thailand) showed the border crossing of the disease through land and air. Sequences had many nucleotide changes compared to the original COVID-19 sequence from Wuhan, China (NC045512.2) [9].

The D614G mutation, found in January 2021, was first detected in January 2020 in China and Germany. It outnumbered the initial coronavirus by April/May [10]. The G614 variant had improved receptor binding and transmission, becoming the dominant global strain [9]. Molecular courting also indicated that this novel mutation ascended early in the pandemic. Genetic drift caused a high rate of a specific variant without selective advantage [11]. The Q677H mutation found in some of our samples might affect the stability of the spike protein as it lay near the furin cleavage site, but there was no evidence of its effect on the pathogenicity [12]. No differences were identified in severity between patients with Q677H and without.

The N501Y mutation observed in Alpha, and L452R and T478K in Delta variant, lied at RBD. In the U.K., South Africa, and Brazil, the N501Y mutation had increased binding affinity to ACE2. L452R showed reduced neutralization from several monoclonal antibodies [13,14,15] and convalescent plasma [13]. L452R independently appeared in different lineages between December 2020 and February 2021. This substitution might result from viral adaptation due to increased immunity in the population [9]. The T478K mutation at the interface complex with human ACE2 might also affect the affinity with human cells and viral infectivity [16]. Kristian Andersen identified a notable feature: polybasic, furin binding at positions 681–688, and subsequent cleavage of the Spike protein worsened its stability. Cleavage of subunits increased binding affinity to the ACE2 receptor markedly [17]. Unfortunately, all Delta and Kappa variants developed with the P681H mutation at FBS and would cause more transmission.

Kappa variants also showed a E484Q mutation at the RBD’s receptor-binding motif (RBM). As the central functional motif, RBM directly binds the human ACE2 receptor [18]. E484N and E484K escape from several monoclonal antibodies and antibodies in convalescent plasma [15,19]. Due to the presence of both L452R and E484Q mutations, our Kappa variant would affect the antigenicity and subsequent immune protection [18,19]. Investigations using pseudoviruses with RBD mutations revealed that the neutralizing activity of plasma from vaccinated people decreased onefold to thrice against N501Y and E484K [20].

Finally, Omicron came up with multiple mutations at RBD and FBS, which would cause the virus to be more transmissible and affect the host’s immune protection. Clinical data showed rapid transmission without evidence of the disease severity and a high mortality rate. K417 is the epitope for class 1 and class 2 antibodies [21], but mutation would affect only class 1 antibody binding. Therefore, the K417 mutation found in this study would have less effect on the polyclonal antibody response, which is responsible for class 2 antibody binding. They were more vulnerable to substitutions including E484K [22]. K417N and K417T had lower ACE2-binding affinity in addition to their antigenic impact [23].

The development of vaccines and drugs reduced disease severity and the spread of SARS-CoV-2. However, selective pressure and viral evolution yield resistant viral variants. The first FDA-approved Paxlovid is the conserved main protease (M^pro^) inhibitor [24]. M^pro^ is a 3-chymotrypsin-like cysteine protease (3CLpro), which digests two large polyproteins (pp1a and pp1ab) to generate proteins critical for virus replication and transcription. It is encoded as nonstructural protein 5 (Nsp5) at open reading frame 1 (ORF1) [25]. Paxlovid is a useful drug in blunting SARS-CoV-2 disease pathogenesis. Nirmatrelvir is the active ingredient in Paxlovid. Ensitrelvir is another M^pro^ inhibitor in clinical-stage development. In M^pro^, P168S mutation is the most commonly seen naturally occurring variant, followed by a single residue deletion, ΔP168. The ΔP168 mutant protease is found to have increased resistance to nirmatrelvir and ensitrelvir, but the P168S variant has no drug-resistant effect [24]. There was no P168 mutation in our study, and other studies also showed M^pro^ as a highly conserved protein relative to the spike protein [25].

The spike glycoprotein is used as an antigen in vaccine development as it binds to the Angiotensin-Converting Enzyme 2 receptor and enters the host cell, which is crucial in the first step of the infection process. Immunization-induced specific antibodies against the receptor binding domain block and prevent virus invasion [26]. However, we have to monitor the spike glycoprotein mutation as its coding region is one of the most variable sites in the genome [27]. The mutations at that place have effects on vaccine efficacy [28].

Currently, we are using Daan Gene and Bioflux RT-PCR and cobas6800 platforms to diagnose COVID-19. Therefore, we checked the targets of China CDC and Charité protocols. China CDC protocols target at the Nucleocapsid coding region and the ORF1ab region. Charité protocols target at the ORF1ab region and the Envelope coding region. Some Delta variants in our study showed nucleotide substitution at the China CDC forward primer binding site at the Nucleocapsid coding region (28881-28902) and Charite forward primer binding site at ORF1b (15431-15452) only. Some Omicron variants showed substitution at the China CDC forward primer binding site at the Nucleocapsid region (28881-28902) and Charite forward primer binding site at the Envelope region (26269-26294) only. These show that our molecular platforms are still effective in diagnosing COVID-19 [29].

Distribution of our sequences across the whole phylogenetic tree constructed with sequences from 15 countries worldwide, inferring globalization and high transmissibility of disease, warned us to keep stricter vigilance of imported cases. Sequence homology with Europe and Malaysia indicated keeping track of disease transmission patterns and ports of entry from these regions. Ongoing transmission from India from the First to the Sixth Run highlighted the need not to emphasize only the eastern border but to exercise more containment strategy at the west.

Multiple variants claimed sustainable plans and preparation for diagnosis, treatment, prevention, and control measures against repeatedly emerging diseases. Genomic data during waves of world outbreaks showed the dominance of the fittest Delta strain over other strains. This domination predicted the inevitable importation of this strain, and we should use this information effectively to prepare for future control measures.

## 5. Conclusions

Detailed information about mutations at receptor binding sites, immune epitopes, and primer binding sites offered important intelligence data to combat the novel virus. We should maintain our efforts in genomic surveillance. Finally, the whole genome sequencing platform in this study opens our capability for diagnosing emerging infectious diseases in the upcoming days in the challenging world. Multiple variants and mutations have necessitated increased vigilance at ports of entry and the development of effective control measures. Genomic surveillance with the observation of evolutionary data is required to predict imminent threats of the current disease and diagnose emerging infectious diseases.

## Figures and Tables

**Figure 1 vaccines-11-00006-f001:**
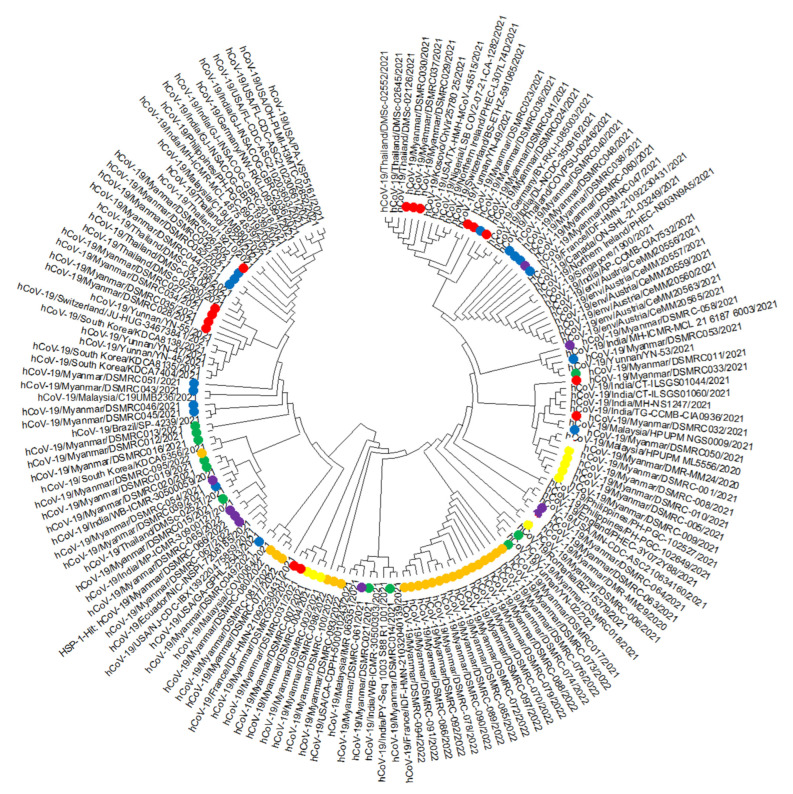
Phylogenetic tree comparing Myanmar SARS-CoV-2 sequences with 78 previously published sequences. Label: Yellow = First Run, Green = Second Run, Red = Third Run, Blue = Fourth Run, Purple = Fifth Run, and Orange = Sixth Run; We performed sequence alignment using MEGA version 7 with the MUSCLE algorithm, neighbor-joining analysis method, and the Kimura 2-parameter model with 1000 bootstrap replications for phylogenetic analysis.

**Table 1 vaccines-11-00006-t001:** Genotyping of Myanmar SARS-CoV-2 sequences with demographic data.

Sequence Name	Age	Sex	Residence	Vaccination	Severity	Travel History	Collection Date (DD-MM-YY)	PANGO Lineage	GISAID Clade	GISAID Accession Number
001/2021	40	Male	NayPYiTaw	No	Mild	Local	04-01-21	B.1.36.16	GH	EPI_ISL_833041
002/2021	37	Male	NayPYiTaw	No	Mild	Local	09-01-21	B.1.36.16	GH	EPI_ISL_849736
004/2021	38	Male	NayPYiTaw	No	Mild	Local	09-01-21	B.1.36.16	GH	EPI_ISL_849737
005/2021	41	Male	NayPYiTaw	No	Mild	Local	09-01-21	B.1.36.16	GH	EPI_ISL_849738
006/2021	42	Male	NayPYiTaw	No	Mild	Local	09-01-21	B.1.36.16	GH	EPI_ISL_849739
007/2021	37	Male	NayPYiTaw	No	Mild	Local	09-01-21	B.1.36.16	GH	EPI_ISL_849740
008/2021	38	Male	NayPYiTaw	No	Mild	Local	09-01-21	B.1.36.16	GH	EPI_ISL_849741
009/2021	41	Male	NayPYiTaw	No	Mild	Local	09-01-21	B.1.36.16	GH	EPI_ISL_849742
010/2021	42	Male	NayPYiTaw	No	Mild	Local	09-01-21	B.1.36.16	GH	EPI_ISL_849743
011/2021	35	Female	Kalay	Not known	Mild	Local	28-05-21	B.1.617.2	GK	EPI_ISL_2612300
012/2021	55	Male	Myeik	Not known	Mild	Local	28-05-21	B.1.1.7	GR	EPI_ISL_2592630
013/2021	22	Female	Myeik	Not known	Mild	Local	28-05-21	B.1.617.1	G	EPI_ISL_2593764
014/2021	51	Female	Myeik	Not known	Mild	Local	28-05-21	B.1.1.7	GR	EPI_ISL_2595725
015/2021	28	Female	Mandalay	Not known	Mild	Local	01-06-21	B.1.617.2	GK	EPI_ISL_2595726
016/2021	7	Female	Mandalay	Not known	Mild	Local	01-06-21	B.1.617.2	GK	EPI_ISL_2595860
017/2021	25	Female	Mandalay	Not known	Mild	Local	01-06-21	B.1.617.2	GK	EPI_ISL_2596341
018/2021	18	Female	Mandalay	Not known	Mild	Local	01-06-21	B.1.617.2	GK	EPI_ISL_2596803
019/2021	23	Male	Tamu	Not known	Mild	Local	02-06-21	B.1.617.1	G	EPI_ISL_2597228
020/2021	42	Male	Tamu	Not known	Mild	Local	02-06-21	B.1.617.1	G	EPI_ISL_2597229
021/2021	42	Male	Yangon	Not known	Mild	Local	26-05-21	B.1.617.1	G	EPI_ISL_2597312
022/2021	43	Male	Nay Pyi Taw	No	Mild	Local	07-07-21	AY.30	GK	EPI_ISL_5424999
023/2021	55	Female	Mandalay	No	Dead	Local	25-06-21	B.1.617.2	GK	EPI_ISL_5427023
024/2021	76	Female	Monywa	No	Severe/ICU	Local	12-08-21	B.1.617.2	GK	EPI_ISL_5427644
026/2021	52	Female	Kalay	Covidshield	Mild	Local	11-08-21	B.1.617.2	GK	EPI_ISL_5427934
027/2021	30	Male	Mawlamyine	Covidshield	Asymptomatic	Local	09-08-21	AY.30	GK	EPI_ISL_5428660
028/2021	28	Male	Lashio	Sinopharm	Mild	Local	11-08-21	B.1.617.2	GK	EPI_ISL_5428815
029/2021	4	Female	Nay Pyi Taw	No	Mild	Local	16-08-21	AY.30	GK	EPI_ISL_5428817
030/2021	31	Male	Nay Pyi Taw	No	Critical/ICU	Local	16-08-21	AY.30	GK	EPI_ISL_5428988
031/2021	46	Female	Sittwe	Sinopharm	Mild	Local	25-08-21	AY.30	GK	EPI_ISL_5428993
032/2021	26	Male	Yangon	Covaxin	Mild	Local	07-06-21	B.1.617.2	GK	EPI_ISL_5428995
033/2021	31	Male	Yangon	Covaxin	Severe/ICU	Local	25-08-21	B.1.617.2	GK	EPI_ISL_5429000
034/2021	45	Female	Nay Pyi Taw	Covaxin	Mild	Local	06-07-21	AY.30	GK	EPI_ISL_5429003
035/2021	37	Female	Taungoo	Covidshield	Mild	Local	05-07-21	B.1.617.2	GK	EPI_ISL_5429005
036/2021	3	Female	Nay Pyi Taw	No	Mild	Local	16-08-21	B.1.617.2	GK	EPI_ISL_5429009
037/2021	33	Male	Mandalay	Covidshield	Mild	Local	25-06-21	AY.30	GK	EPI_ISL_5429011
038/2021	34	Female	Taunggyi	Covidshield	Mild	Local	23-11-21	B.1.617.2	GK	EPI_ISL_7337304
039/2021	38	Male	ThinganNyinaung	Covidshield	Mild	Thailand	12-11-21	AY.85	GK	EPI_ISL_7337866
040/2021	29	Male	Mawlamyine	Covidshield	Mild	Thailand	03-11-21	AY.85	GK	EPI_ISL_7337871
041/2021	20	Male	Kengtung	Covidshield	Mild	Laos	08-11-21	B.1.617.2	GK	EPI_ISL_7337873
042/2021	21	Female	Kengtung	Covidshield	Mild	Laos	08-11-21	B.1.617.2	GK	EPI_ISL_7338123
043/2021	39	Male	Shan	Covidshield	Mild	Local	24-11-21	AY.79	GK	EPI_ISL_7338109
044/2021	34	Male	Naypyitaw	Covidshield	Mild	Local	01-12-21	B.1.617.2	GK	EPI_ISL_7338111
045/2021	36	Male	Naypyidaw	Covidshield	Mild	Local	03-12-21	AY.129	GK	EPI_ISL_7338113
046/2021	47	Female	Naypyitaw	Covidshield	Mild	Local	03-12-21	AY.98	GK	EPI_ISL_7338114
047/2021	52	Male	Yangon	Covidshield	Mild	Local	30-11-21	AY.129	GK	EPI_ISL_7338119
048/2021	25	Male	Yangon	Covidshield	Mild	Malaysia	01-12-21	B.1.617.2	G	EPI_ISL_7338201
049/2021	36	Male	Yangon	Covidshield	Mild	Malaysia	01-12-21	AY.114	G	EPI_ISL_7338374
050/2021	54	Male	Mawlamyine	Covidshield	Mild	Thailand	04-12-21	AY.59	GK	EPI_ISL_7338375
051/2021	17	Female	Mawlamyine	Covidshield	Mild	Thailand	04-12-21	AY.59	GK	EPI_ISL_7338376
052/2021	18	Male	Mandalay	Covidshield	Mild	Local	03-12-21	B.1.617.2	GK	EPI_ISL_7338402
053/2021	54	Male	Mandalay	Covidshield	Mild	Local	03-12-21	B.1.617.2	GK	EPI_ISL_7338766
054/2021	52	Male	Tachileik	Not known	Mild	Thailand	06-12-21	AY.129	GK	EPI_ISL_8920487
058/2021	65	Male	Kalay	Not known	Mild	Local	27-12-21	B.1.617.2	GK	EPI_ISL_8920488
060/2021	38	Male	Yangon	Not known	Mild	Local	27-12-21	AY.79	GK	EPI_ISL_8920490
061/2021	43	Male	Yangon	Not known	Mild	Local	27-12-21	AY.79	GK	EPI_ISL_8920491
063/2021	29	Male	Yangon	Not known	Mild	UAE	30-11-21	BA.1	GRA	EPI_ISL_8920492
064/2021	29	Male	Yangon	Not known	Mild	UAE	30-11-21	BA.1	GRA	EPI_ISL_8920493
065/2022	4	Female	Yangon	Not known	Mild	Philippines	04-01-22	BA.2.3	GRA	EPI_ISL_8920494
066/2022	44	Male	Yangon	Not known	Mild	Philippines	04-01-22	BA.2.3	GRA	EPI_ISL_8920501
067/2022	33	Male	Yangon	Not known	Mild	Malaysia	04-01-22	BA.1.1	GRA	EPI_ISL_8920500
070/2022	42	Male	Nay Pyi Taw	Covaxin	Mild	Local	12-03-22	BA.2	GRA	EPI_ISL_11149371
072/2022	34	Male	Nay Pyi Taw	Covaxin	Mild	Local	12-03-22	BA.2	GRA	EPI_ISL_11149374
073/2022	28	Male	Myanaung	Not known	Mild	Local	14-03-22	BA.2	GRA	EPI_ISL_11149375
074/2022	43	Female	Nay Pyi Taw	Covaxin	Mild	Local	14-03-22	BA.2	GRA	EPI_ISL_11149531
076/2022	36	Female	Nay Pyi Taw	Sinopharm	Mild	Local	14-03-22	BA.2	GRA	EPI_ISL_11149533
077/2022	30	Male	Taungdwingyi	Covidshield	Mild	Local	14-03-22	BA.2	GRA	EPI_ISL_11149535
078/2022	55	Female	Taungoo	Sinopharm	Mild	Local	14-03-22	BA.2.10.1	GRA	EPI_ISL_11149537
079/2022	49	Male	Nay Pyi Taw	Covaxin	Mild	Local	14-03-22	BA.2	GRA	EPI_ISL_11149539
080/2022	30	Female	Nay Pyi Taw	Covidshield	Mild	Local	14-03-22	BA.2	GRA	EPI_ISL_11149543
081/2022	23	Female	Nay Pyi Taw	Covidshield	Mild	Local	14-03-22	BA.2	GRA	EPI_ISL_11149547
085/2022	27	Female	Nay Pyi Taw	Covaxin	Mild	Local	11-03-22	BA.2	GRA	EPI_ISL_11149552
086/2022	57	Female	Thayet	Covaxin	Mild	Local	11-03-22	BA.2	GRA	EPI_ISL_11149554
088/2022	32	Female	Nay Pyi Taw	Sinopharm	Mild	Local	11-03-22	BA.2.10	GRA	EPI_ISL_11149559
089/2022	29	Female	Nay Pyi Taw	Sinopharm	Mild	Local	11-03-22	BA.2.10.1	GRA	EPI_ISL_11149562
090/2022	32	Female	Homalin	Covidshield	Mild	Local	19-02-22	BA.2.10.1	GRA	EPI_ISL_11149564
091/2022	26	Male	Homalin	Covidshield	Mild	Local	19-02-22	BA.2	GRA	EPI_ISL_11149614
092/2022	17	Female	Homalin	Covidshield	Mild	Local	19-02-22	BA.2.10	GRA	EPI_ISL_11149615
093/2022	60	Female	Homalin	Covidshield	Mild	Local	19-02-22	BA.2.10.1	GRA	EPI_ISL_11149617
094/2022	56	Male	Homalin	Covidshield	Mild	Local	19-02-22	BA.2	GRA	EPI_ISL_11149639
095/2022	17	Male	Homalin	Covidshield	Mild	Local	19-02-22	BA.2	GRA	EPI_ISL_11149640
097/2022	42	Male	Nay Pyi Taw	Covidshield	Mild	Local	11-03-22	BA.2	GRA	EPI_ISL_11149643
098/2022	42	Male	Nay Pyi Taw	Covaxin	Mild	Local	11-03-22	BA.2.38	GRA	EPI_ISL_11149644
100/2022	34	Female	Nay Pyi Taw	Covaxin	Mild	Local	11-03-22	BA.2.10	GRA	EPI_ISL_11149739

## Data Availability

Data are available at the corresponding GISAID accession number in Table 1.

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
