# Peer review of "Genomic Tracking of SARS-CoV-2 Variants in Myanmar"

_vaccines, 2022, doi:10.3390/vaccines11010006_

Round 1

Reviewer 1 Report

The manuscript is well written and clear for the reader. It provides a series of useful information on the evolution of SARS-CoV-2 variants in Myanmar between the beginning of 2021 and the first part of 2022. 

There are some minor revisions:

Figure 1: Describe the software and the parameters you used to build the phylogenetic tree in the caption of the figure.  

Line 172-180: Make more clear the sense of this sentence in the context of your discussions. I didn't well understand the final meaning of the description of the platform you used in this part of the discussion.  

Move the period of the lines 193-198 (Detailed information...) to the beginning of the paragraph "Conclusions".

Author Response

For Reviewer 1:

Thank you very much for your comments.

We added the name of the software and the parameters used to make alignment and build the phylogenetic tree in the caption of the figure.

We made line 172-180 to become clearer.

We moved the lines 193-198 to the beginning of the paragraph “Conclusions”           

We would be too much grateful if you can check again. Thank you very much in advance.

Reviewer 2 Report

The manuscript on "Genomic tracking of SARS-CoV-2 variants in Myanmar", submitted to the Journal Vaccines provides information on new mutation across SARS Variant at receptor binding sites, immune epitopes, and primer binding sites based on genome sequencing data. This information is extremely important for development of novel diagnostic assays for SARS, identifying the cause of viral transmission and novel targets for vaccine development. The manuscript is well written with clarity of results and the data is of high importance for SAR-CoV2 research. I would like to recommend this article for publication after revision in abstract and discussion.

Comments: In the abstract and discussion, the author's states the importance of the research for diagnostic purposes and effective control measures. It will be important to add statement in abstract and elaborate in discussion on how this information on mutation in SARS structural proteins can help identifying novel targets for vaccine development and improve the therapeutic strategy for new variants of SAR-CoV-2. 

Author Response

For Reviewer 2:

Thank you very much for your comments.

We added statement in abstract about importance of information on mutation on vaccine and therapeutic strategy. We also elaborated in discussion about them with additional 2 paragraphs.

We also revised English a little bit according to recommendation in email.

We would be too much grateful if you can check again. Thank you very much in advance.